# The Complex Transcriptional Landscape of Magnetosome Gene Clusters in *Magnetospirillum gryphiswaldense*

Marina Dziuba,a,b Cornelius N. Riese,a Lion Borgert,a* Manuel Wittchen,c,d Tobias Busche,d Jörn Kalinowski,d René Uebe,a Dirk Schülera

aDepartment of Microbiology, University of Bayreuth, Bayreuth, Germany

bInstitute of Bioengineering, Research Center of Biotechnology of the Russian Academy of Sciences, Moscow, Russia

cGerman Network for Bioinformatics Infrastructure, Bielefeld, Germany

dCenter for Biotechnology, University of Bielefeld, Bielefeld, Germany

Marina Dziuba and Cornelius N. Riese contributed equally to this work. Author order was determined alphabetically.

**ABSTRACT** Magnetosomes are complex membrane organelles synthesized by magnetotactic bacteria (MTB) for navigation in the Earth's magnetic field. In the alphaproteobacterium *Magnetospirillum gryphiswaldense*, all steps of magnetosome formation are tightly controlled by >30 specific genes arranged in several gene clusters. However, the transcriptional organization of the magnetosome gene clusters has remained poorly understood. Here, by applying Cappable-seq and whole-transcriptome shotgun RNA sequencing, we show that *mamGFDCop* and *feoAB1op* are transcribed as single transcriptional units, whereas multiple transcription start sites (TSS) are present in *mms6op*, *mamXYop*, and the long (>16 kb) *mamABop*. Using a bioluminescence reporter assay and promoter knockouts, we demonstrate that most of the identified TSS originate from biologically meaningful promoters which mediate production of multiple transcripts and are functionally relevant for proper magnetosome biosynthesis. In addition, we identified a strong promoter in a large intergenic region within *mamXYop*, which likely drives transcription of a noncoding RNA important for gene expression in this operon. In summary, our data suggest a more complex transcriptional architecture of the magnetosome operons than previously recognized, which is largely conserved in other magnetotactic *Magnetospirillum* species and, thus, is likely fundamental for magnetosome biosynthesis in these organisms.

**IMPORTANCE** Magnetosomes have emerged as a model system to study prokaryotic organelles and a source of biocompatible magnetic nanoparticles for various biomedical applications. However, the lack of knowledge about the transcriptional organization of magnetosome gene clusters has severely impeded the engineering, manipulation, and transfer of this highly complex biosynthetic pathway into other organisms. Here, we provide a high-resolution image of the previously unappreciated transcriptional landscape of the magnetosome operons. Our findings are important for further unraveling the complex genetic framework of magnetosome biosynthesis. In addition, they will facilitate the rational reengineering of magnetic bacteria for improved bioproduction of tunable magnetic nanoparticles, as well as transplantation of magnetosome biosynthesis into foreign hosts by synthetic biology approaches. Overall, our study exemplifies how a genetically complex pathway is orchestrated at the transcriptional level to ensure the balanced expression of the numerous constituents required for the proper assembly of one of the most intricate prokaryotic organelles.

**KEYWORDS** MTB, *Magnetospirillum*, magnetosomes, operons, promoters, transcription, transcriptome

Address correspondence to Dirk Schüler, dirk.schueler@uni-bayreuth.de.

* Present address: Lion Borgert, Department of Biochemistry and Molecular Biology, University Hospital of Bonn, Bonn, Germany.

The research reveals a previously unappreciated complexity of transcriptional architecture of biosynthetic operons controlling the assembly of magnetosomes, one of the most intricate prokaryotic organelles.

One of the most complex organelles found in prokaryotic cells is the magnetosome, which serves in magnetotactic bacteria (MTB) as a sensor for navigation in the Earth's magnetic field (1). In the long-standing model organism *Magnetospirillum gryphiswaldense* strain MSR-1 (referred to here as MSR-1) and related MTB, magnetosomes consist of a monocrystalline core of magnetite ($Fe_3O_4$) enclosed within a membrane. The unprecedented crystalline and magnetic properties of bacterial magnetosomes make them highly attractive in several biotechnical and biomedical settings, such as magnetic imaging and hyperthermia, as well as magnetic separation and drug targeting (2). Their application potential can be further enhanced by genetic or chemical coupling of functional moieties to the magnetosome membrane (3). Furthermore, it has been suggested to build magnetic nanostructures within eukaryotic cells for local heat generation or as reporters for magnetic imaging by borrowing genetic parts from bacterial magnetosome biosynthesis in the field of "magnetogenetics" (4, 5).

In MSR-1, biosynthesis of magnetosomes proceeds in several steps, including (i) invagination of the cytoplasmic membrane to form a magnetosome membrane (MM) vesicle; (ii) sorting and dense packing of specific magnetosome proteins (MAP) into the MM; (iii) uptake of iron and biomineralization of well-ordered magnetite crystals; and (iv) the assembly and positioning of nascent magnetosomes into linear chains (6–8). Besides some functions contributed by generic metabolic pathways (9), all these processes are governed by more than 30 proteins designated as Mam (magnetosome membrane), Mms (magnetosome particle membrane-specific), and Feo (magnetosome-specific $Fe^{2+}$ transport system), which together constitute a sophisticated machinery exerting strict control over each step of magnetosome biosynthesis. In MSR-1, all MAPs are encoded within five major polycistronic operons (MagOPs, Fig. 1A) as follows: *mamABop* (16.4 kb), *mamGFDCop* (2.1 kb), *mms6op* (3.6 kb), *mamXYop* (5 kb), and *feoAB1op* (2.4 kb) (10–13). The MagOPs are clustered within an ~110-kb chromosomal region termed the genomic magnetosome island (MAI), where they are interspersed with genes irrelevant for magnetosome biosynthesis (14–17). The long *mamABop* comprises 17 genes and encodes all the essential factors for magnetosome biosynthesis, whereas the other four operons play important but accessory roles in magnetite biomineralization, chain assembly, and its intracellular positioning (10, 17, 18). Transfer and expression of all five MagOPs from MSR-1 caused magnetosome biosynthesis in two different nonmagnetic bacteria, further substantiating the key roles of this gene set in the process (19, 20). However, several further attempts to transplant magnetosome biosynthesis to other bacteria have so far failed, partly owing to the poor and imbalanced transcription from the as-yet-uncharacterized native promoters (Dziuba and Schüler, unpublished).

In order to build such an intricate organelle, the MAPs have to be properly expressed and targeted to the MM in defined and highly balanced stoichiometries that range, for example, from 2 (e.g., MamX and MamZ) to 120 copies (Mms6) per magnetosome particle (21), which requires a precise control over expression. One fundamental layer of regulation is expected to act at the level of gene transcription, which has been addressed by only few studies so far. Schübbe et al. demonstrated by reverse transcription-PCR (RT-PCR) that genes from the three magnetosome gene clusters known at the time, *mamABop*, *mms6op*, and *mamGFDCop*, are cotranscribed and thus represent genuine operons in MSR-1 (11). Additionally, the study also identified a single transcription start site (TSS) for each transcript by primer extension analysis, which suggested that each operon is transcribed as a single unit (TU) driven by a primary promoter residing upstream of the first gene of each operon. (11). Although no additional promoters could be identified within the operons in that study, the presence of internal promoters, especially in *mamABop* (16.4 kb), could not be ruled out based on the data available at that time (11). Later, the activity of a primary promoter (P*mamY*) upstream of the newly discovered *mamXYop* was demonstrated by a green fluorescent protein (GFP) reporter, whereas no additional promoters were identified (12). In *feoAB1op*, a primary promoter (P*feoA1*) was revealed by a LacZ reporter gene fusion in MSR-1 (13). Despite that magnetosomes are synthesized only within a narrow range of growth conditions, i.e., microoxic to anaerobic and in the presence of sufficient iron (22, 23),

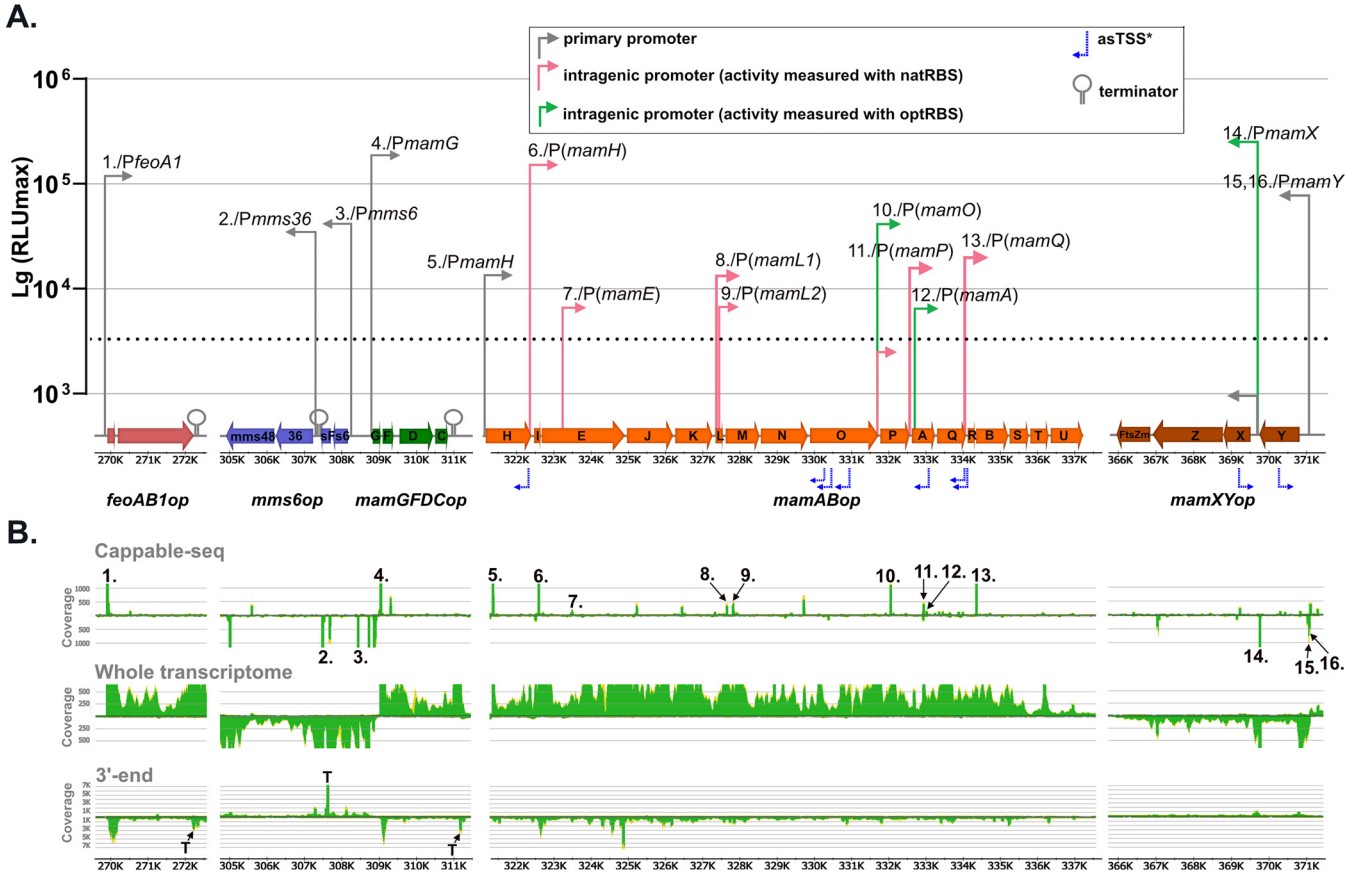

**FIG 1** Molecular organization and transcriptional architecture of the MagOPs revealed in this study. (A) Position of the promoters, whose activities were confirmed by the bioluminescence assay, terminators, and asTSS in the MagOPs. Arrow height indicates the promoter strength measured by luminescence (see the text for details). A slash separates a TSS number designation and the corresponding promoter (as in Table S2 in the supplemental material). (B) Localization of TSS and TTS predicted by the transcriptome data sets. Numbers indicate TSS as in Table S2. "T" in the 3′-end panel indicate TTS.

magnetosome genes have been found to be mostly constitutively expressed, where growth conditions only weakly affected the abundance of magnetosome proteins, as demonstrated by quantitative reverse transcription-PCR (qRT-PCR), Western blotting (11), and transcriptome analysis (Riese et al., in preparation).

While these previous studies seemed to indicate a rather simple transcriptional organization of magnetosome genes, a growing amount of data suggest that a large fraction of operons in other prokaryotes are complex, i.e., contain more than one internal promoter, terminator, or both, and hence are transcribed as mutually overlapping TUs (24–27). For instance, studies on the 14-kb photosynthetic gene cluster in purple nonsulfur *Proteobacteria* and the 27-kb *fla/che* cluster in *Bacillus subtilis* (28, 29), suggested that an intricate landscape of transcriptional regulatory elements may be common to such long polycistronic operons. Understanding of the transcriptional organization of MagOPs in comparable detail is not only essential for unraveling the magnetosome biosynthesis regulation, but also for its future reconstitution, engineering, and tuning by synthetic biology approaches in homologous and heterologous hosts. All of this prompted us to reassess the architecture of the major magnetosome operons in MSR-1 by a comprehensive approach that included various RNA sequencing techniques, bioluminescence reporter assays, and promoter knockouts. By this, we confirmed the activity of the primary promoters suggested before and revealed multiple novel promoters within the MagOPs. We further show that these internal promoters can drive expression of downstream genes in the absence of primary promoters. Taken together, our data suggest a much more complex transcriptional organization of the MagOPs than deemed before and thus contribute to unveiling the fundamentals of magnetosome biosynthesis.

## RESULTS

**Identification of putative TSS and TTS in magnetosome operons by RNA sequencing.**
Transcription start sites (TSS) were determined across the genome by the Cappable-seq
technique and whole transcriptome shotgun sequencing (WTSS). Cappable-seq is a
method of enriching for the 5′ end of primary transcripts by enzymatically tagging the
triphosphorylated 5′ end of RNA, which enables the determination of TSS at single-base
resolution (30). In addition, transcription termination sites (TTS) were determined using 3′
end sequencing, by unambiguous peaks in combination with a read coverage decrease in
the WTSS data set (31). The identified TSS were classified into four groups using an auto-
mated script: (i) primary TSS (pTSS, i.e., positioned in front of the coding sequence), (ii) intra-
genic TSS in sense (iTSS) or (iii) antisense orientation (asTSS), and (iv) other TSS (oTSS) (Fig.
S1 in the supplemental material). From a total of 9,639 TSS identified in the entire transcrip-
tome, 319 were found in the MAI (position bp 269182 to 371200 in the genome), and 77
within the MagOPs (Summarized in Table S2 and Fig. 1B). Similar to the previously reported
prevalence of intragenic TSS in bacterial and archaeal transcriptomes (25, 30), the majority
(69.3%/6,674 TSS) of the TSS defined across the genome of MSR-1 occur within coding
sequences, with 3,273 TSS (34.0%) in sense orientation (iTSS), 3,401 TSS (35.3%) in antisense
orientation (asTSS), and 319 (3.3%) classified as others (oTSS) (Fig. S1). Only 2,646 (27.4%)
represented primary TSS (pTSS). The distribution of TSS within the MAI and the MagOPs was
largely similar (Fig. S1), with a total of 23 pTSS (25.0%), 39 iTSS (42.4%), and 27 asTSS (29.3%)
identified within the MagOPs. For enhancement of the TSS detection specificity, we
increased the enrichment score threshold to 1.4 and compared the putative TSS to the other
RNA-seq data sets (see Materials and Methods), resulting in 8 pTSS (32%), 7 iTSS (28%), 9
asTSS (36%), and 1 oTSS (4%) within the MagOPs (positions and number designations are
shown in Table S2 and Fig. 1B).

Among the MagOPs, *feoAB1op* and *mamGFDCop* appear to have a canonical structure,
with only a single pTSS located immediately upstream of the first gene of each operon (TSS
1 and TSS 4), but no internal TSS were found. Transcriptional terminations within the last
180 bp of *feoB* (*feoAB1op*) and 10 bp downstream of *mamC* (*mamGFDCop*) were detected in
the 3′ end data set.

In *mms6op*, a single pTSS (TSS 3) was detected 346 nucleotides (nt) upstream of *mms6*. In
addition, another unambiguous pTSS (TSS 2) is present within the intergenic 175-bp region
between *mmsF* and *mms36*. Furthermore, a putative TTS immediately downstream of *mmsF*
was found, whereas no TTS was determined after *mms48* in the 3′ end sequencing data set.
These observations indicate that *mms6op* might be transcribed as two separate TUs, *mms6-
mmsF* and *mms36-mms48*, each driven by its own TSS and separated by a terminator.

In *mamXYop*, two pTSS were located upstream of *mamY* (TSS 15 and TSS 16). An
additional pTSS (TSS 14) was found 102 bp upstream of *mamX*. The presence of a pro-
moter in this region was hypothesized previously, but could not be confirmed by a
GFP reporter assay (12). Besides, two asTSS were identified at positions 369,133 bp and
370,214 bp within *mamXYop*. The read coverage in the WTSS data set showed steady
transcription throughout the complete operon, gradually declining at the end of *ftsZm*,
but with no unambiguous TTS suggested by the 3′ end sequencing (Fig. 1B).

Although the single pTSS (TSS 5), which was found 17 bp upstream of *mamH*, the
first gene in *mamABop*, did not exceed the thresholds applied for TSS identification, it
was added since its position is associated with the promoter "P*mamAB*" (referred to as
P*mamH* in this study) determined in the previous studies (11, 32). Furthermore, eight
iTSS were detected within the coding sequences of *mamH* (TSS 6), *mamE* (TSS 7),
*mamL* (TSS 8), *mamO* (TSS 10), *mamP* (TSS 11), *mamA* (TSS 12), and *mamQ* (TSS 13).
Additionally, a second putative iTSS (TSS 9) in *mamL*, which was identified by a con-
spicuous rise in read coverage in the WTSS data set, was further investigated. In addi-
tion, seven asTSS with significant read coverage in the Cappable-seq as well as the
WTSS data set were detected within *mamH*, *mamO*, *mamA*, and *mamQ*. The asTSS at
the position 330,492 bp was assigned due to the overlapping read coverage to the
neighboring asTSS (330,355 bp), despite being below the applied threshold of 1.4.

Sequencing of 3′ ends revealed no distinct TTS within or at the end of the operon. Although a conspicuous increase in the 3′ end sequencing read coverage was observed within *mamE*, this was not accompanied by a decrease of read coverage in the downstream genes in the WTSS data. The continuous read coverage of the *mamABop* in WTSS argues for its uninterrupted transcription and the possible generation of at least one single long transcript, as suggested by Schübbe et al. (11). However, the presence of multiple additional TSS within *mamABop* implies the existence of several overlapping TUs along with this potential long transcript.

**Evaluation of predicted TSS by luminescence reporter assay.** Next, we wanted to experimentally verify the predicted sense TSS and estimate the activities of the potential corresponding promoters. Bacterial luminescence was chosen as a reporter because of its extremely high sensitivity in comparison to fluorescence and chromogenic reporters previously used in magnetospirilla (33–35). The maximal value of normalized light units (RLU$_{max}$) was used to compare the relative strength of the tested promoters (Fig. 1A, Table S3). By precise chromosomal integration of all cassettes into the *attTn7* site by Tn7 transposition (R. Uebe, manuscript in preparation), we aimed to eliminate potential positional effects (Fig. 2A). Two terminator sequences, tr2 of phage lambda and *rrnB* T1 from *Escherichia coli*, were inserted immediately upstream of the promoter of interest (POI) to insulate it. However, preliminary tests of promoterless (P-less) control cassettes revealed a weak (3,213.64 ± 496.32 RLU$_{max}$) but detectable light signal (Fig. 2Bi). This was likely caused by transcriptional activity of the neighboring promoter(s) and indicated that the efficiency of termination by tr2 and T1 in MSR-1 was much lower than in *E. coli*, in which it can approach 100% (36). Nonetheless, tests of several clones containing the control cassette demonstrated that this activity remained roughly identical in at least three independent experiments. Therefore, these signals were treated as background that would be predictably reproduced in all measured promoters, and only those POI that exceeded the RLU$_{max}$ of the P-less control were assumed to be active promoters (Fig. 2Bi to 2Bvii).

Reporter fusions exhibited significant transcriptional activity for all tested TSS, confirming that they are generated by genuine promoters. Thus, the activities of P*feoA1* and P*mamG* ranged from 100,592.9 to 143,000.8 RLU$_{max}$ and 131,925.3 to 325,856.8 RLU$_{max}$, respectively (Table S3). This result was consistent with previous observations of high activities estimated by a GFP reporter for P*mamG* and *lacZ* for P*feoA1* (13, 34). In *mms6op*, both putative promoters associated with TSS 2 and TSS 3 (P*mms6* and P*mms36*, respectively) exhibited significant activities: 29,347.87 to 36,089.24 RLU$_{max}$ for P*mms36* and 35,956.18 to 66,616.58 RLU$_{max}$ for P*mms6*. This substantiates the putative existence of two bicistronic mRNAs, as suggested by RNA sequencing.

In *mamXYop*, the primary promoter P*mamY* generated 69,670.43 to 88,076.17 RLU$_{max}$. Previously, P*mamY* was estimated to exhibit only 22.5% of P*mamG* activity by GFP and GusA reporters (12). Here, the use of bioluminescence revealed a slightly higher, but still comparable, activity of approximately 35.3% of P*mamG*. Previously, fusion of the intergenic fragment between *mamY* and *mamX* (P*mamX*) to a GFP reporter failed to reveal promoter activity (12). Consistently, we were unable to detect any activity of this region with our bioluminescence reporter, even when up to 20 nt from the 5′ end of the *mamX* coding sequence was included in the leader (data not shown). Inspection of this region did not reveal any sequence resembling a canonical ribosome binding site (RBS) (5′-AGGAGA-3′) between −5 to −10 nt ahead of the start codon of *mamX* (Table S4). However, when the fusion construct was augmented by insertion of the optimized Shine-Dalgarno sequence (optRBS, see Materials and Methods, [34]), strong light emission became detectable (174,463.3 to 337,600.0 RLU$_{max}$). This confirmed the high transcriptional activity of this region as predicted by the transcriptomic data but suggested that translation is inefficient due to the absence of a native RBS and, hence, P*mamX* might rather generate an as-yet-unidentified species of noncoding RNA (ncRNA) in its native context.

The predicted primary promoter P*mamH* of the *mamABop* demonstrated relatively weak but significant activity (10,467.7 to 23,270.1 RLU$_{max}$). The activity of the potential

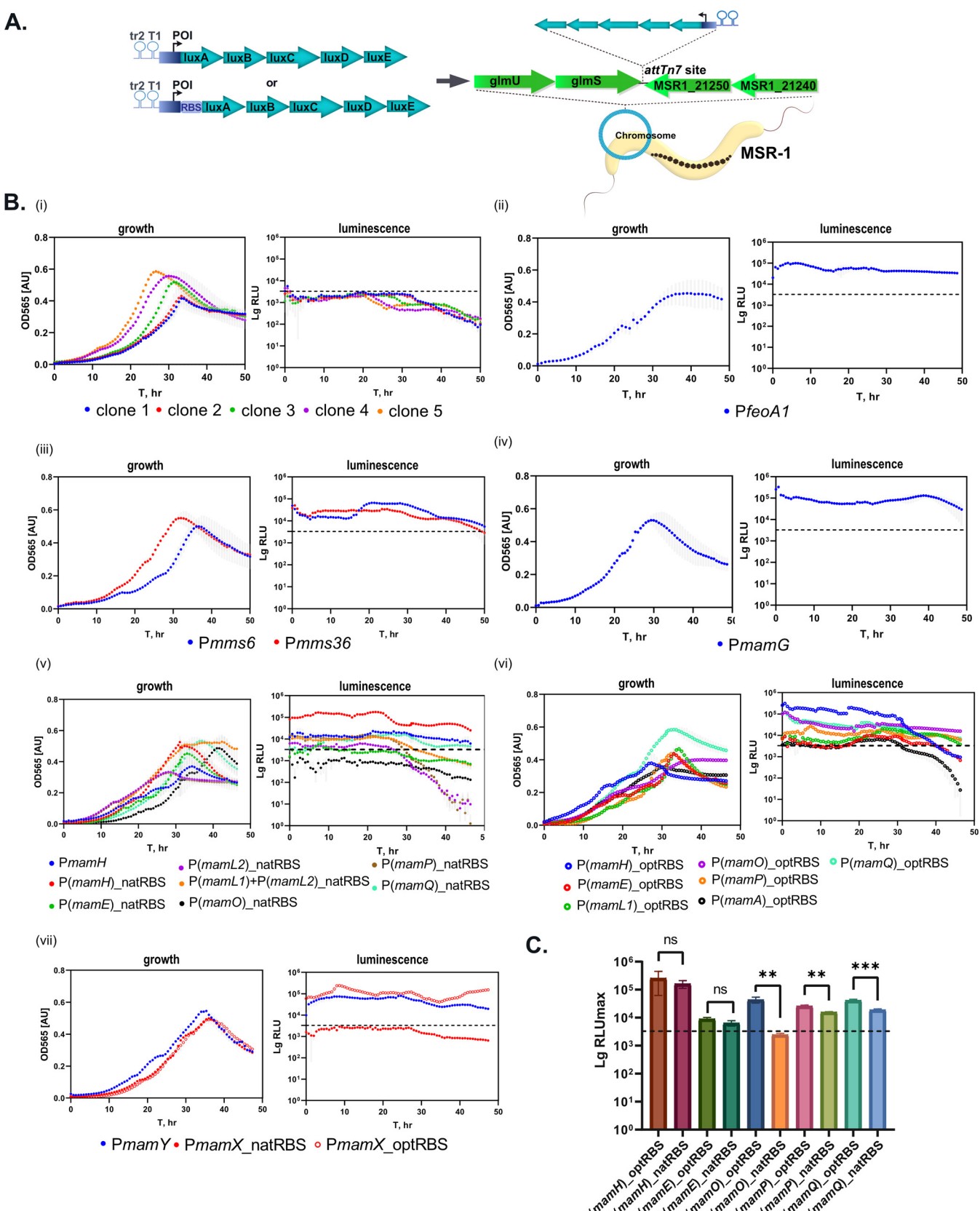

**FIG 2** Activity of promoters from the MagOPs evaluated by the bioluminescence assay. (A) Schematic representation of the cloning strategy for the *in vivo* measurement of the promoter activity. Tr2 and T1, terminators; POI, promoter of interest. (B) Growth and luminescence curves of representative clones:

intragenic promoters corresponding to the predicted TSS 6 to 11 and TSS 13 (Table S2), i.e., P(*mamH*), P(*mamE*), P(*mamL1*), P(*mamL2*), P(*mamO*), P(*mamP*), and P(*mamQ*), were first measured with their native leader sequences. Indeed, inspection of the regions directly upstream of the next genes immediately following each intragenic promoter revealed sequences that may function as an RBS for translation of the *mamI*, *mamJ*, *mamM*, and *mamA* products (hereafter referred to as natRBS [native RBS]). However, no natRBS close to the start codons of *mamP*, *mamQ*, or *mamR* could be predicted with confidence (Table S4). Nonetheless, all promoter regions were cloned according to the same procedure, i.e., with the leader sequence spanning to the start codon of the next downstream gene.

Within *mamL*, iTSS 8 and 9 were found separated by 161 nt, suggesting that two different promoters reside within the gene, which, however, have overlapping leader sequences with a shared natRBS upstream of *mamM*. Therefore, a longer sequence comprising both promoters, P(*mamL1*) + P(*mamL2*), and a shorter sequence harboring only the putative second promoter (P*mamL2*), were individually fused to the *luxAE* reporter.

When tested with their potential natRBS, P(*mamH*), P(*mamL1*)+P(*mamL2*), P*mamP*, and P*mamQ* demonstrated significant activity, whereas the signals generated from P(*mamE*) and P(*mamL2*) were very weak, and no activity above the background could be detected for P(*mamO*) (Fig. 1A, Fig. 2Bv). Among these promoters, P(*mamH*) demonstrated the highest $RLU_{max}$, ranging from 117,963.0 to 215,346.9, which is approximately 75% of the P*mamG* activity. The activity of P(*mamA*) with the natRBS was not estimated; however, it exceeded the threshold signal when cloned with the optRBS (see below). In summary, we confirmed transcriptional activity for most of the tested intragenic promoters which was also coupled to translation of the bioluminescence reporter, likely due to the presence of natRBS in the leader sequences of the corresponding transcripts. This also suggests that multiple mRNAs are likely produced within the *mamABop*.

The activity of the predicted intragenic promoters was also evaluated after augmenting the sequences with optRBS. This allowed us to estimate the activity of the promoters independent of the efficiency of naturally occurring RBS. The activity of P(*mamH*) and P(*mamE*) measured with optRBS did not differ significantly from the natRBS, whereas the light emission with optRBS was enhanced approximately 1.5-fold in P(*mamP*), and 2-fold in P(*mamQ*) (Fig. 2C). This was likely caused by different ribosome-binding efficiencies of the natRBS in comparison to the optRBS. Interestingly, P(*mamO*) did not cause any significant bioluminescence when cloned with its native leader, but demonstrated considerable activity with optRBS, ranging from 35,344.6 to 55,643.9 $RLU_{max}$. As in the case of P*mamX*, this correlates with the absence of a canonical RBS in the putative leader downstream of the iTSS within *mamO* (Table S4) and implies the lack of efficient translation despite the significant transcriptional activity. Similarly, this suggests that an ncRNA might be generated from this promoter. In addition, the use of optRBS allowed us to independently measure the activity of P(*mamL1*), which reached up to 22,164.3 $RLU_{max}$, and the activity of a putative promoter corresponding to TSS 12, P(*mamA*), which demonstrated only weak activity ranging from 5,741.4 to 6,874.2 $RLU_{max}$.

**Exploration of the newly identified promoters *in vivo* by promoter knockout.** Next, we asked whether the promoters revealed within the magnetosome operons can drive transcription of downstream genes independently of the primary promoters located immediately upstream of their operons. In this case, one would expect that inactivation of P*mamH*, P*mms6*, and P*mamY* will not completely abolish transcription of the corresponding operons, resulting in weaker phenotypes resembling the Δ*mamH*, Δ*mms6*Δ*mmsF*, and Δ*mamY* mutants (10, 12, 18). On the contrary, if P*mamH*, P*mms6*, and P*mamY* are the

**FIG 2** Legend (Continued)
(i) P-less control. Five clones are shown to demonstrate reproducibility of the maximal background light signal that was used as a threshold for all the subsequent measurements; (ii) *feoAB1op*; (iii) *mms6op*; (iv) *mamGFDCop*; (v) *mamABop*, with native RBS (natRBS); (vi) *mamABop*, with optimized RBS (optRBS); (vii) *mamXYop*. Dotted line indicates the background activity derived from the $RLU_{max}$ of the P-less control. Standard deviations are shadowed in gray. (C) Comparison of the maximal RLU ($RLU_{max}$) generated by the tested promoters with their native RBS (natRBS) with those augmented with the optimized RBS (optRBS). Statistical significance was estimated using the *t* test. Asterisks indicate the points of significance, **, *P* value < 0.01; ***, *P* value < 0.001.

only or main promoters driving the transcription of the entire operons, their elimination would result in significantly more severe impairments of magnetosome formation, likely phenocopying ΔmamABop, Δmms6op, and ΔmamXYop deletions, respectively (10, 17). Likewise, by knockout of Pmms36 and PmamX, the phenotypes of Δmms36Δmms48 and ΔmamXΔmamZΔftsZm, respectively, would be expected (10, 12, 37). To test this hypothesis, promoter knockouts were generated by replacing the promoter-comprising sequences by an artificial promoter-free sequence (PFS) of equal length (except PmamY, see Materials and Methods) (Fig. 3A).

Elimination of the primary promoter PmamH resulted in a mutant (ΔPmamH) forming smaller (26.9 ± 8.3 nm versus 32.3 ± 10.5 nm in the wild type [WT]) and fewer magnetosomes in comparison to the WT (Fig. 3B and C), but not with complete absence of magnetosomes, as in ΔmamABop (10). Instead, the phenotype of ΔPmamH was virtually identical to that previously described for ΔmamH, in which the magnetosome size and number were also significantly reduced (12). This suggests that only mamH was silenced by the PmamH knockout, whereas transcription of the remaining 16 genes of mamABop was still driven by intragenic promoters. Since mamH is immediately followed by the essential magnetosome genes mamI and mamE, whose deletion entirely eliminates magnetosome formation (10), this implies that their expression has to be mediated primarily by P(mamH) (TSS 6). Consistently, complementation of the ΔPmamH with PmamH-mamH in trans essentially restored the magnetosome diameter and number to WT levels (Fig. 3C, Fig. S2).

In ΔPmms6, neither the magnetosome number nor magnetic response of cells was affected (Fig. 3B, Fig. S3), whereas magnetosomes appeared to be smaller than in the WT (29.9 ± 9.8 nm versus 32.3 ± 10.5 nm in the WT) (Fig. 3B and 3Ci). This moderate decrease in the size reproduced the phenotype of the Δmms6ΔmmsF mutant, but was unlike the more severe decrease in magnetosome size and number that was described for the mutant lacking the entire mms6op (10). Complementation of ΔPmms6 with Pmms6-mms6-mmsF restored the magnetosome size back to the WT level (Fig. 3C, Fig. S2). Furthermore, elimination of Pmms36 resulted in a significantly reduced magnetosome number in comparison to the WT, which also could be restored by complementation with Pmms36-mms36-mms48 (Fig. 3B and C, Fig. S2). Taken together, these results suggest that transcription of mms36 and mms48 is primarily driven by Pmms36. Notably, the phenotypic effect of simultaneous silencing of mms36 and mms48 was different from their individual deletions, which had previously demonstrated enlarged magnetosomes with up to a 10 to 30% increase in the average diameter (10).

The ΔPmamY strain demonstrated the characteristically displaced magnetosome chains of ΔmamY in 77% of analyzed cells, which also correlated with a reduced cellular magnetic response (Cmag 1.01 ± 0.09, Fig. S3) (18). Under standard conditions, the cells had an inconsistent phenotype, with most cells having regular magnetosomes and a minor proportion containing aberrant flake-like magnetosomes, reminiscent of the mutant with the entire mamXYop eliminated (Fig. 3B) (17). It has been demonstrated that the formation of flake-like magnetosomes observed after the individual deletions of mamX, mamZ and ftsZm is more pronounced under nitrate-deprived conditions, likely due to the shared redox control over the biomineralization by their products and the denitrifying enzymes (12, 37). Therefore, to check whether PmamY drives transcription of the entire mamXYop operon, we grew ΔPmamY cells under microoxic conditions in a medium in which sodium nitrate was replaced by an equimolar amount of ammonium chloride. As expected, under these conditions, ΔPmamY mutants showed severely impaired biomineralization and the displaced magnetosome chains (Fig. 4), suggesting that elimination of PmamY affects transcription of the entire operon. It also indicates that, unlike the additional promoters in mms6op and mamABop, the intergenic promoter PmamX does not compensate for the absence of PmamY. Under standard conditions, cells of ΔPmamX were virtually indistinguishable from the WT with respect to magnetic response, magnetosome biomineralization, or chain organization (Fig. 3B, Fig. S3). However, cultivation of them with ammonium resulted in flake-like magnetosomes as in ΔPmamY (Fig. 4). This implies that both PmamY and PmamX are required for proper expression of mamX-mamZ-ftsZm genes.

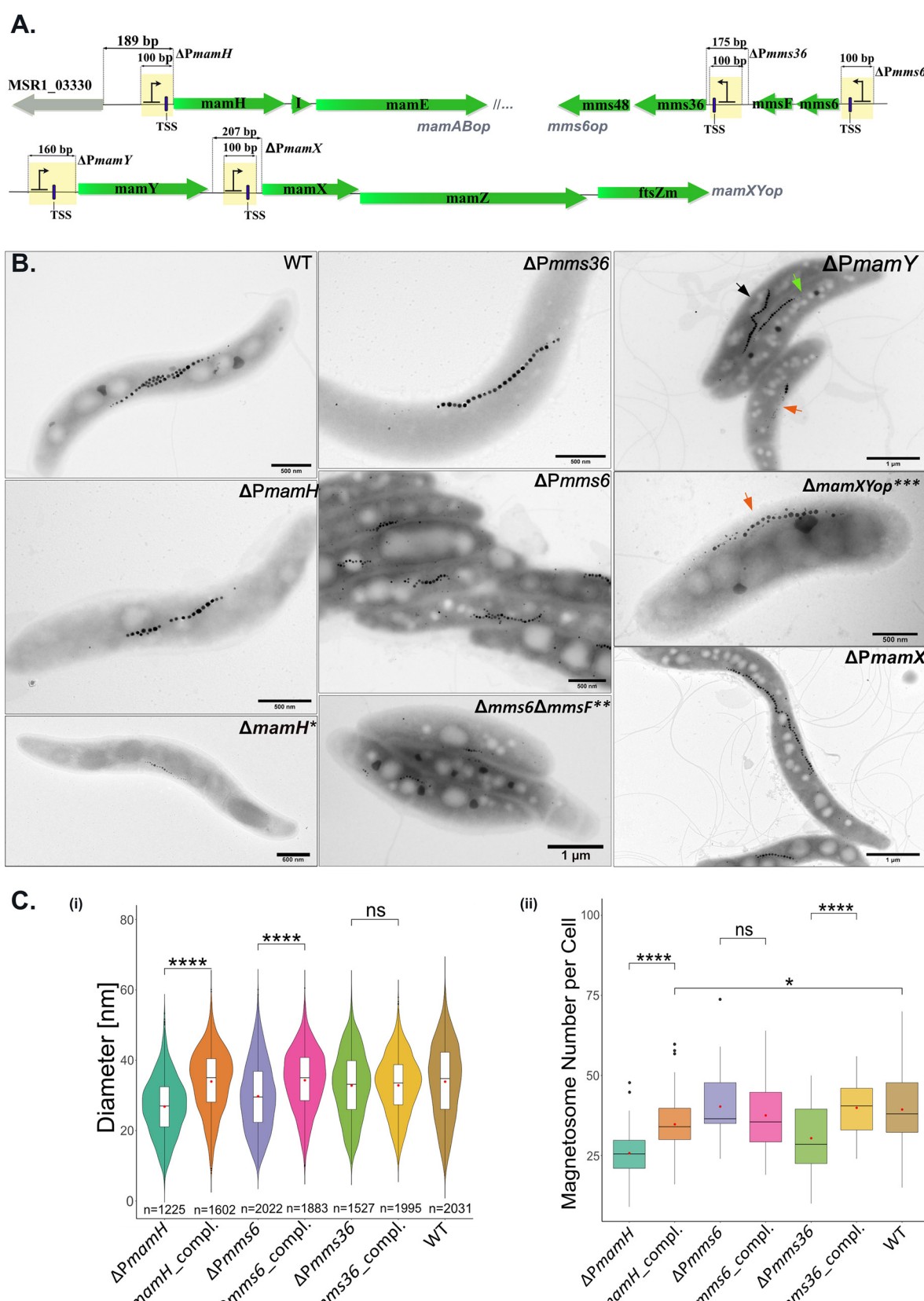

**FIG 3** Exploration of the newly identified promoters in vivo by promoter knockout. (A) Schematic representation of the mutagenesis strategy. Yellow bars indicate the regions that were replaced with the promoter-free sequences (PFS). (B) TEM micrographs of the

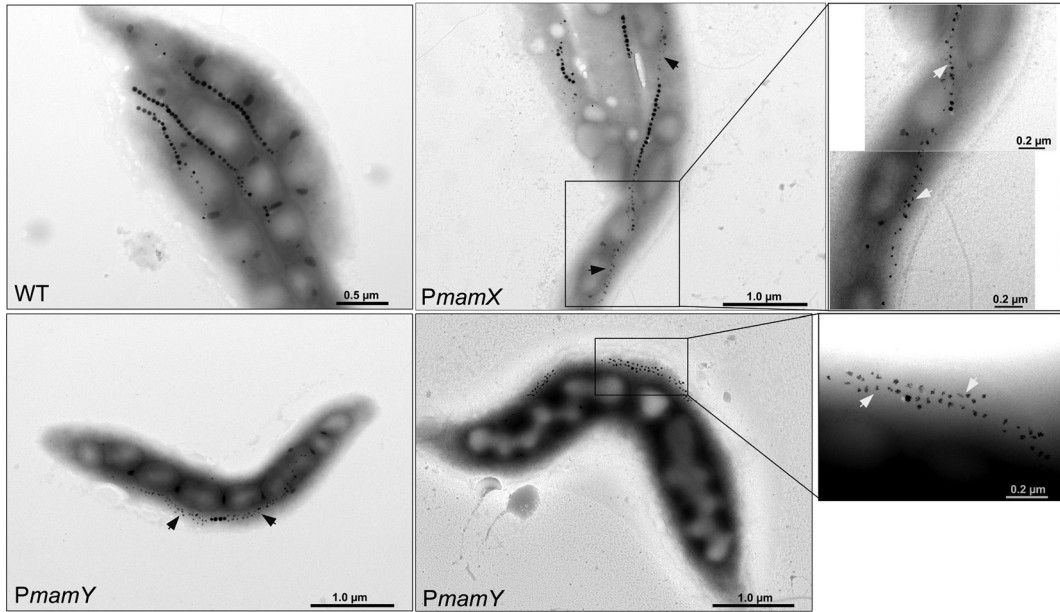

**FIG 4** TEM micrographs of the ΔP*mamY* and ΔP*mamX* mutants grown under nitrate deprivation. Two representative cells of the ΔP*mamY* are shown. Arrows indicate flake-like magnetosomes.

The complementation of ΔP*mamY* with P*mamY-mamY* restored the regular chain position. However, frequent flake formation was still observed, suggesting a lack or low expression of *mamX*, *mamZ*, and *ftsZm* (Fig. S2). Interestingly, complementation of ΔP*mamX* with P*mamX-mamXZftsZm* essentially restored the WT-like appearance of the magnetosomes observed in cells cultivated with ammonium (Fig. S2). Hence, the result reinforces that P*mamX* can modulate the expression of the *mamX-mamZ-ftsZm* genes in addition to P*mamY*, although the exact role of the generated transcript is not clear. Since *mamX*, *mamZ*, and *ftsZm* seem to be especially important for magnetosome biomineralization under nitrate deprivation conditions, we next tested whether the activity of either P*mamY* or P*mamX* is regulated by nitrate. To this end, bioluminescence was measured in clones harboring P*mamY* and P*mamX* fused to *luxABCDE* in the absence or presence of nitrate. However, no significant difference in light emission was detected (Fig. S4), suggesting that the activity of these promoters is not regulated in response to nitrate deprivation.

**Promoter sequences within magnetosome operons are conserved across *Magnetospirillum* spp.** The complex landscape of transcription initiation in the MagOPs revealed in MSR-1 raised the question of whether such an organization is significant for proper magnetosome formation. If so, it would be expected to be conserved to a certain degree across different species. In other MTB, genes associated with magnetosome biosynthesis are also found in operon-like gene clusters (38–42). Although the gene content and order vary between different taxonomic lineages, some magnetosome genes have higher synteny rates even in distantly related groups (38, 39). Interestingly, in many cases, the first gene in syntenic gene groups is an orthologue of the gene found to comprise a functional promoter

**FIG 3** Legend (Continued)
promoter mutants grown in the standard medium. TEM micrographs of the Δ*mamH*, Δ*mms6*Δ*mmsF*, and Δ*mamXYop* published previously are shown for comparison (*, from reference 12 [© John Wiley & Sons Ltd., reproduced with permission]; **, republished from reference 10; ***, republished from *PLoS One* [17]). In ΔP*mamY*, three typical cell types occurring in the population are indicated by arrows: black, cells with magnetosome chains indistinguishable from the WT; green, a chain mispositioned to the geodetic line within the cells; orange, magnetosome chains with prevailing flake-like magnetosomes. (C) (i) Violin plots displaying magnetosome diameter in the mutants in which the promoters were substituted with PFS and the corresponding complemented mutants. Numbers of the measured particles are indicated at the bottom of each violin plot. (ii) Boxplots demonstrating the magnetosome number per cell in the promoter substitution and complemented mutants. Significance values were calculated by Kruskal-Wallis test; ****, *P* value of less than 0.0001; ns, not significant. Boxplots display the minimum, maximum, and median of each data set. Red points indicate mean. At least 50 cells were measured for each strain.

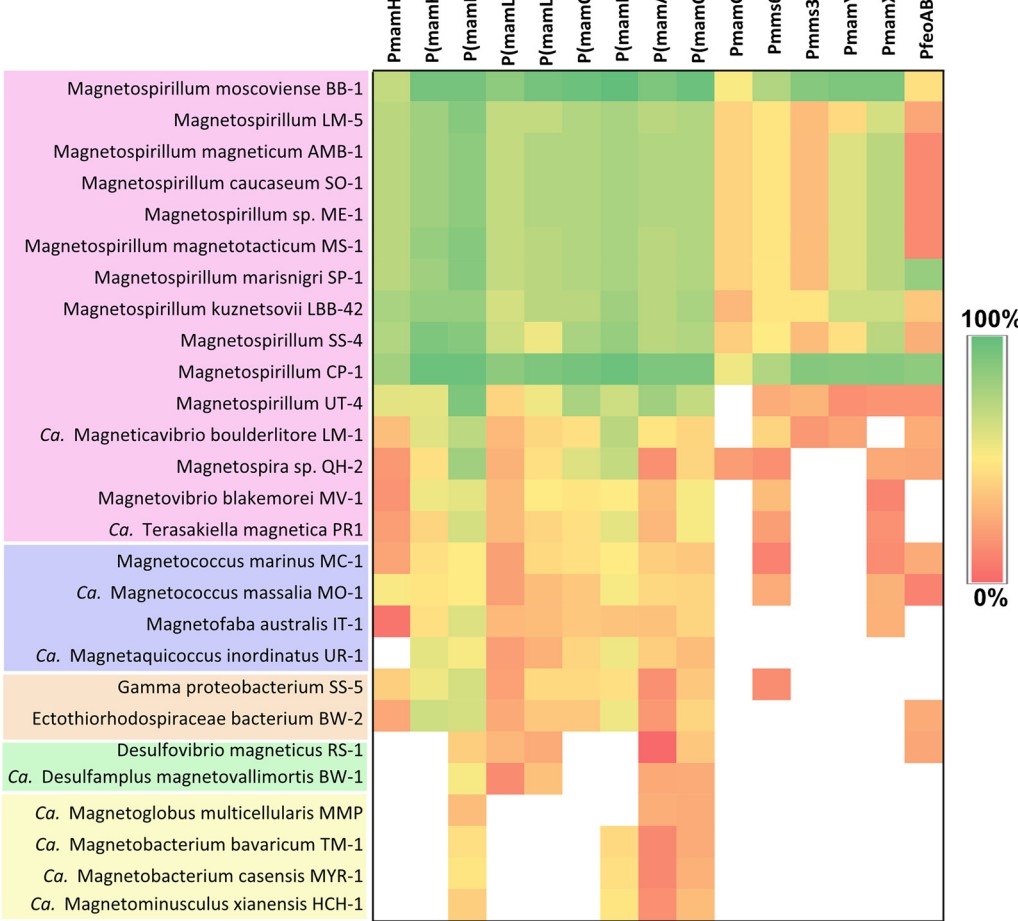

**FIG 5** Conservation of the promoter sequences from magnetosome operons as identified in MSR-1 across MTB (see the text for details). White regions indicate that the region is not found in a genome.

in the current research. For instance, *mamL* is followed by *mamM* in many phyla in which these genes are present (except *Nitrospirae* and *Omnitrophica*), and the order *mamQ-(mamR)-mamB* is preserved in *Nitrospirae*, *Nitrospinae*, and *Proteobacteria*. This prompted us to estimate the sequence conservation of the promoter-containing regions in various phylogenetic groups of MTB (Fig. 5). To this end, sequences positioned −5 to −50 to a confirmed TSS were extracted from promoters tested in MSR-1 and compared to the orthologous sequences in other species.

As expected, sequence conservation was highest across *Magnetospirillum* species, where high similarity was found for P*mamH*, P*mms6*, P*mamY*, P*mamX*, and all intragenic promoters, whereas P*mamG*, P*mms36*, and P*feoAB1* regions were more variable. This implies functional conservation of most of the promoters and, in general, similar organization of transcriptional landscape in MagOPs for different magnetospirilla. A notorious exception is *Magnetospirillum* strain UT-4, in which only several intragenic promoter sequences, P(*mamE*), P(*mamO*), P(*mamP*), P(*mamA*), P(*mamQ*), were conserved. This correlates with the distant, ancestral position of the magnetosome genes from this strain to other known *Magnetospirillum* spp. (42). Although relatively high similarity of the sequences orthologous to promoters within *mamE* and *mamP* were found in alphaproteobacterial MTB, in general, the promoters from the MagOPs were not conserved outside of *Magnetospirillum* spp.

## DISCUSSION

By combination of various techniques, we were able to map multiple TSS within the magnetosome operons with high precision, evaluate the transcriptional activity of the corresponding promoters, and estimate their function in magnetosome biosynthesis.

The results suggested that *mamGFDCop* and *feoAB1op* are organized as classic polycistronic operons, in which transcription is driven by a single conventional promoter and intercepted by a terminator at the 3′ end. The other three operons turned out to have a more complex transcriptional landscape.

One of the key findings of this study is the discovery of multiple promoters residing within the coding sequences of the long *mamABop* operon. Knockout of P*mamH*, the primary promoter of this operon, had only a minor effect on magnetosome formation and silenced only the gene that is located immediately downstream of it, i.e., *mamH*, suggesting that P*mamH* is not essential for transcription of the major part of *mamABop*. At the same time, the transcription of the following genes, including all the essential ones in the operon, were maintained by the intragenic promoters. One of the most crucial internal promoters must be P(*mamH*), as the downstream genes (*mamI* and *mamE*) are essential for magnetosome formation. Moreover, the reporter assay demonstrated that P(*mamH*) is one of the most active promoters among the ones measured in the current study and the strongest in *mamABop*, with the activity exceeding that of the primary promoter P*mamH* ~10 times. Interestingly, the unintended elimination of P(*mamH*) concurrent with the deletion of *mamH* by Raschdorf et al. did not entirely abolish magnetosome biosynthesis, but only caused the formation of fewer and smaller particles (12). This effect was attributed to the absence of *mamH*, suggesting that the primary promoter P*mamH* can also drive low-level transcription of following genes in the absence of the intragenic P(*mamH*), thus supporting the existence of a long polycistronic transcript, as previously suggested (11). Nonetheless, complementation with *mamH* in *trans* only partially restored the magnetosome size and number in the Δ*mamH* mutant according to Raschdorf et al. (12), whereas complementation of the P*mamH* knockout mutant in this study restored the magnetosome size to the WT levels (12). Therefore, the weaker activity of P*mamH* putatively compensates the lack of P(*mamH*) only to some extent, which emphasizes the importance of the latter for proper transcription of the essential magnetosome genes in *mamABop* (12). To our knowledge, this is the first demonstration that intragenic promoters can exceed primary promoters in activity and potentially play a major role in driving expression of large operons.

The adaptive role of multiple transcripts generated in the *mamABop* is not yet clear. On the one hand, the multiple promoters residing within the 16-kb *mamABop* operon might compensate the potential instability of the single long mRNA by splitting the operon into several smaller TUs, thus making the transcription of the whole operon more efficient. On the other hand, this might represent one of the mechanisms to ensure a certain stoichiometric ratio of gene products required for the proper assembly of the magnetosome organelle. As we found no obvious correlation between the MAP abundance (21) and the promoter strengths defined either by Cappable-seq scores or $RLU_{max}$ (data not shown), the highly divergent copy numbers of MAPs are likely to be further regulated at the translational level. This has been shown to be largely independent of the growth conditions, suggesting that the promoters within the operons, like the primary promoters of the MagOPs, are unlikely to be subjected to any conditional regulation.

*Mms6op* comprises two TUs, *mms6-mmsF* and *mms36-mms48*, separated by a terminator and each driven by a separate promoter. The presence of a terminator does not mean *per se* that the TUs are independent, since the transcriptional readthrough due to the inefficient termination can still occur, and hence their transcription can be coupled (24, 43). Interestingly, in all known magnetotactic *Magnetospirillum* species, *mms36* and *mms48* are always preceded by *mms6-mmsF*, suggesting that this coupled organization might be preserved by natural selection.

We also revealed a very active additional promoter within the *mamXYop* (P*mamX*). However, the lack of reporter expression in the absence of optRBS strongly argues against translation of the produced transcript in the native context, including potential leaderless translation (44). Nonetheless, the knockout of P*mamX* resulted in the

production of aberrant flake-like magnetosomes under nitrate deprivation, which implies that the expression of one or all of *mamX*, *mamZ*, and/or *ftsZm* was affected and, hence, P*mamX* activity is necessary for proper magnetosome biosynthesis under these conditions. At the same time, the adverse effect of this deletion on magnetosome formation was compensated by transfer of P*mamX-mamXZftsZm*. The lack of translation of the generated transcript from P*mamX* on the one hand, and its functional importance on the other hand, suggests that it might represent a noncoding RNA (ncRNA) with a potential regulatory function. However, identification of the exact type and characteristics of the produced RNA species will require further experimentation.

Comparison of the promoter sequences in various MTB suggests that the transcriptional organization of *mamABop* and *mamXYop* revealed in MSR-1 is conserved across the species of *Magnetospirillum*. A notable exception was strain UT-4, in which the promoter sequences were the least conserved. According to Monteil et al. (42), UT-4 possesses the magnetosome genes that are ancestral for *Magnetospirillum*, whereas the evolutional history of magnetosome operons in other *Magnetospirillum* strains was shaped by repeated loss and regain by horizontal transfer. Therefore, the transcriptional organization of magnetosome operons as in MSR-1 likely evolved not in the common ancestor of magnetospirilla, but after their speciation. At the same time, the lack of conservation outside *Magnetospirillum* suggests independent evolution of transcriptional regulation of the MagOPs in different phylogenetic groups. This is a plausible scenario, considering the long evolutionary distances between the MTB genomes and the evidence that new promoters can evolve rapidly (45, 46).

Besides shedding light on the mechanisms underlying control over magnetosome formation, the insights into the transcriptional architecture of the MagOPs obtained in this study have several important practical implications. First, the high-resolution map of transcription initiation will enable synthetic biology approaches to transcriptionally engineer the magnetosome operons for enhanced and controlled magnetosome production, e.g., through replacing the native promoters of individual transcriptional units by stronger and tunable promoters. Likewise, data gained in this study will facilitate the rational design of synthetic versions of magnetosome operons optimized for the expression in foreign organisms (19, 20), as poor transcription of native magnetosome clusters has proven to be one of the key hurdles for successful transplantation of magnetosome biosynthesis to different bacteria. Second, our study provides a catalog of well-characterized promoters with different strengths for constructing expression cassettes in magnetospirilla and other *Alphaproteobacteria*. In conclusion, our study unveils how a genetically complex pathway is orchestrated at the transcriptional level to ensure the proper assembly of one of the most intricate prokaryotic organelles.

## MATERIALS AND METHODS

**Bacterial strains and culture conditions.** If not specified otherwise, *Magnetospirillum gryphiswaldense* strain MSR-1 (DSM 6361) (47, 48) was routinely cultivated in flask standard medium (FSM, 10 mM HEPES [pH 7.0], 15 mM potassium lactate, 4 mM NaNO$_3$, 0.74 mM KH$_2$PO$_4$, 0.6 mM MgSO$_4$·7H$_2$O, 50 $\mu$M iron citrate, 3 g/liter soy peptone, 0.1 g/liter yeast extract), in flasks containing 2% (vol/vol) O$_2$ in the headspace, at 120 rpm agitation (49). Selection for the mutants was carried on solid FSM with 1.5% (wt/vol) agar and 5 $\mu$g/ml kanamycin (Km).

*E. coli* WM3064 strains carrying plasmids were cultivated in lysogeny broth (LB) supplemented with 0.1 mM DL-$\alpha$,$\varepsilon$-diaminopimelic acid (DAP) and 25 $\mu$g/ml Km at 37°C, with 180 rpm agitation. Characteristics of the strains used in this study are summarized in Table 1.

**RNA isolation, library preparation, sequencing, and mapping to the reference genome.** Transcription initiation, expression coverage, and transcription termination were investigated by Cappable-seq, whole-transcriptome shotgun sequencing (WTSS), and 3′ end sequencing, respectively. For RNA isolation, cells of MSR-1 were cultivated in 5-liter screw-cap bottles at 25°C. Cells were harvested at mid-growth phase (optical density at 565 nm [OD$_{565}$] = 0.2) by centrifugation at 8,300 $\times$ *g* and 4°C for 10 min using a Sorvall RC-5B Plus centrifuge (Thermo Fisher Scientific, Waltham, USA) and flash frozen with liquid nitrogen prior to total RNA isolation. Magnetosome biosynthesis was verified using magnetically induced differential light scattering method C$_{mag}$ as described previously (50) and transmission electron microscopy (TEM). RNA isolated from biological duplicates using the mirVana RNA isolation kit (Thermo Fisher Scientific, Waltham, USA) was treated by DNase, checked by capillary electrophoresis, pooled together, and subsequently used for all library preparations and sequencing by Vertis Biotechnologie AG (Freising, Germany).

**TABLE 1** Bacterial strains and vectors used in this work

| Strain or vector | Characteristics/application | Source/reference |
|---|---|---|
| **Strains** | | |
| *Magnetospirillum gryphiswaldense* MSR-1 | WT, archetype | Lab collection, DSM 6361 |
| *E. coli* WM3064 | *thrB1004 pro thi rpsL hsdS lacZΔM15 RP4-1360 Δ(araBAD) 567 ΔdapA1341::[erm pir]*. Donor strain for transformation by conjugation, $\alpha,\varepsilon$-diaminopimelic acid (DAP) auxotroph. | William Metcalf, UIUC, unpublished |
| **Vectors** | | |
| pBamII-Tn7-P-luxAE | *KmR, AmpR, p15A ori, Tn7, tr2, T1, luxABCDE;* a plasmid for the transcriptional fusion of a promoter (P) and the *lux* operon. Suicide vector, a cassette is introduced by chromosomal insertion mediated by Tn7 into the attTn7 site. | This work |
| pORFM-galK | *KmR, npt, galK, tetR, mobRK2;* general vector for GalK counterselection | 55 |
| pBamII-Tn5 | *KmR, AmpR, p15A ori, mini-Tn5;* general vector used for complementation experiments. Suicide vector, a cassette is introduced by random chromosomal insertion mediated by mini-Tn5 | Uebe, manuscript in preparation |

For the enrichment of primary 5′ ends, a modified version of the Cappable-sequencing technique was used (30). Briefly, 5′ triphosphorylated RNA was capped with 3′-desthiobiotin-TEG-guanosine 5′ triphosphate (DTBGTP) (New England BioLabs, Ipswich, MA, USA) using the vaccinia capping enzyme (New England BioLabs, Ipswich, MA, USA). The biotinylated RNA was then enriched by reversible binding to a streptavidin column, followed by washing and elution of the 5′ fragments. The uncapped control was also applied to the streptavidin column to control for unspecific binding to the column matrix. Afterward, adapter ligation, reverse transcription, and amplification of the cDNA were performed according to the instructions for the TrueSeq Stranded mRNA library (Illumina, San Diego, USA) for both libraries. Single-end sequencing for the two libraries was performed on an Illumina NextSeq 500 system using 1 × 75 bp read length.

For the WTSS library, rRNA was depleted from the pooled RNA sample using the Ribo-Zero rRNA removal kit for bacteria (Illumina, San Diego, CA, USA). The remaining mRNA was purified using the Agencourt AMPure XP kit (Beckman Coulter Genomics, Chaska, MN, USA) and analyzed by capillary electrophoresis. Fragmentation of mRNA, reverse transcription, adapter ligation, and PCR amplification were performed according to the TrueSeq Stranded mRNA library instructions (Illumina). Single-end sequencing was performed on an Illumina NextSeq 500 system using 1 × 75 bp read length.

For the 3′ end library preparation, a 3′ Illumina sequencing adapter was ligated to the 3′-OH ends of the rRNA-depleted RNA sample prior to reverse transcription, cDNA fragmentation, sequencing adapter ligation, and cDNA purification using the Agencourt AMPure XP kit (Beckman Coulter Genomics, Chaska, MN, USA). The paired-end sequencing of the PCR amplified cDNA fragments was performed on an Illumina NextSeq 500 system using 2 × 75 bp read length.

The sequencing reads of the four library preparations were trimmed for sequencing adapters as well as low-quality bases prior to mapping to the *M. gryphiswaldense* genome (accession no. CP027526) using the CLC Bio's Genomic Workbench software package (Qiagen, Venlo, Netherlands).

**Annotation of TSS and TTS.** TSS were automatically detected using the Cappable-tools with standard parameters as previously described (30). Briefly, for each position in the genome, the read coverage was normalized to the sequencing depth, resulting in the relative read score (RRS). For TSS identification, the enrichment score was calculated according to the formula enrichment score = $\log_2(RRS/RRS_{control})$, where $RRS_{control}$ is the relative read score in the control library for the same position as in the TSS-enriched library. When the enrichment score was above 1, a putative TSS was annotated. Subsequently, TSS classification was performed based on the localization of the TSS relative to the genome annotation using an in-house script. Subsequently, the putative TSS were curated manually by comparison of read coverage of the TSS to the background, as well as by applying an enrichment score of 1.4 as a threshold. Afterward, the filtered TSS were then evaluated by comparing the putative TSS with the coverage of the other transcriptome sequencing (RNA-seq) data sets using the software ReadXplorer for visualization (62). At least one of the following criteria had to be met for assigning a confident TSS: (i) a read coverage increase in the WTSS data set downstream and (ii) a 3′-end enrichment upstream of the putative TSS. In the cases of TSS 5 and TSS 9 (Fig. 1A), although a conspicuous rise in read coverage could be detected manually in both Cappable-seq and WTSS, they did not pass the applied threshold. Nonetheless, since the promoter associated with TSS 5 (P*mamH*) had already been identified in previous research (11, 32), and TSS 9 could be easily identified in the Cappable-seq and WTSS data sets by manual curation, both TSS were included in the subsequent experimental evaluation.

TTS were manually identified by a significant increase in read coverage above a threshold of 2,500 uniquely mapped reads in the 3′ end sequencing data set combined with a decrease in WTSS coverage up to 150 bp downstream of a coding sequence.

**Molecular and genetic techniques.** Oligonucleotides applied in this study are listed in Table S1 in the supplemental material. To verify and measure the activity of promoters, regions of varying lengths (Table S3) from maximal +450 bp to −112 bp relative to the predicted TSS were PCR amplified and cloned by NdeI and XhoI restriction sites into a suicide vector pBamII-Tn7-P-luxAE (Table 1) upstream of the *Photorhabdus luminescens luxABCDE* operon, which was cloned from pAH328 (51, 52). The vector enables precise and orientation-specific genomic integration of the expression cassette into the *attTn7* site downstream of *glmS* gene by means of the Tn7 transposase (53), (Uebe, manuscript in preparation). Integration of the cassette in the *attTn7* site was verified by PCR with specific primers.

The promoters P*mamH*, P*mms6*, and P*mamY* were inactivated by replacing 100-bp regions (except P*mamY*, where a 160-bp fragment was exchanged) located immediately upstream of the start codon with an inert artificial sequence of equal length that was free of any regulatory elements (the "promoter-free sequence" [PFS]). In case of the intergenic P*mms36* and P*mamX*, the regions upstream of the −20 bp position to the start codon were replaced with the PFS, to keep putative natural RBSs. Maintaining the native sequence lengths was important to avoid potential effects caused by shorter distances to the neighbor promoters located upstream or with altered gene expression due to the reduced leader length.

The PFS (5′-CATTACTCGCATCCATTCTCAGGCTGTCTCGTCTCGTCTCGCTGGGAGTTCGTAGACGGAAACAA ACGCAGAATCCAAGCGCACTGAAGGTCCTCAATCG-3′) was designed as a concatenate of the unique nucleotide sequences UNS1, UNS2, and the first 20 nt of UNS3 that were used previously to generate regulation signature-free homology arms for Gibson assembly (54). The oligonucleotide was inserted by overlapping PCR between two 1- to 1.3-kb sequences flanking the target promoters. The resulting PCR products were phosphorylated by T4 polynucleotide kinase and blunt ligated into the vector for homologous recombination (pORFM-GalK) digested with EcoRV (55). The plasmids were transferred into the wild-type MSR-1 by conjugation, as described elsewhere (56). Selection, counterselection, and screening of the deletion mutants were performed essentially as described previously (55). For genetic complementation, the silenced genes and the corresponding missing promoters were inserted randomly into the mutant chromosome by Tn5 transposition. To this end, the P*mamH-mamH*, P*mms6-mms6-mmsF*, P*mms36-mms36-mms48*, P*mamY-mamY*, and P*mamX-mamX-mamZ-mamFtsZm* regions were PCR amplified from the MSR-1 WT genomic DNA (gDNA), digested with XhoI/BamHI, PacI/BamHI, or XhoI/SacI and ligated into a vector derivate of pBam1, pBamII (57), (Uebe, manuscript in preparation). Positive clones were selected by Km resistance and screened by PCR.

**Luminescence measurements.** At least three randomly selected transconjugants harboring vector pBamII-Tn7-P-luxAE were analyzed in three biological replicates for luminescence. The luminescence signal was detected as arbitrary light units by a multiwell plate reader equipped with a luminometer module (Tecan Infinite M200 PRO) during growth of the cultures in FSM at 28°C and 280 rpm, every 20 min over 200 cycles (72 h). Arbitrary light units were normalized to optical density measured at the wavelength of 565 nm ($OD_{565}$) to obtain relative light units, according to the formula:

$$RLU = \frac{Light\ AU}{OD565\ AU}$$

Maxima of the RLU curves ($RLU_{max}$) were used to compare promoter activities.

**Transmission electron microscopy (TEM).** Cells were concentrated from 2 to 3 ml of culture by centrifugation, adsorbed onto carbon-coated copper grids, and washed twice with deionized water. Samples were imaged with a JEOL-1400 Plus TEM (Japan) at 80 kV acceleration. Micrographs were analyzed with tools implemented in the ImageJ software (58).

**Analysis of promoter sequence conservation.** For each TSS identified by Cappable-seq with the luminescence-confirmed promoter activity, 300 nt upstream of the TSS, a leader sequence and a gene positioned next to TSS, were extracted. Regions homologous to the extracted ones were identified in the genomes of other MTB by blastp (59) of the gene product amino acid sequence (E value cut-off threshold $10^{-5}$) and inspected manually. Homologous DNA sequences were aligned by MAFFT with the default parameters (60) and the sequence identity between the regions aligned to the fragment positioned −5 to −50 to the TSS in MSR-1 were calculated.

**Statistical analysis.** Statistical analysis was carried out by R version 3.6.1 (http://www.r-project.org). Significance in comparison of magnetosome size and number was estimated by Kruskal-Wallis test. Violin and box plots were created using the following R packages: ggplot (https://CRAN.R-project.org/package=ggplot2), ggpubr (https://CRAN.R-project.org/package=ggpubr), dplyr (https://CRAN.R-project.org/package=dplyr), and EnvStats (https://cran.r-project.org/web/packages/EnvStats/index.html). The bioluminescence and growth curves were plotted using GraphPad Prism software (v. 6.01 for Windows).

**Data availability.** The data discussed in this publication have been deposited in NCBI's Gene Expression Omnibus (61) and are accessible through GEO Series accession number GSE168986.

## SUPPLEMENTAL MATERIAL

Supplemental material is available online only.

**FIG S1**, TIF file, 0.4 MB.

**FIG S2**, TIF file, 6.9 MB.

**FIG S3**, TIF file, 0.8 MB.

**FIG S4**, TIF file, 0.4 MB.

**TABLE S1**, PDF file, 0.1 MB.
**TABLE S2**, PDF file, 0.1 MB.
**TABLE S3**, PDF file, 0.1 MB.
**TABLE S4**, PDF file, 0.2 MB.

## ACKNOWLEDGMENTS

This study was supported by the European Research Council (ERC) under the European Union's Horizon 2020 research and innovation program (grant no. 692637 to D.S.).

We are grateful to Bachelor students Filiz Kuybu, Tobias Göttsche, and Katharina Volk for technical assistance. We also thank Stefan Geimer and Rita Grotjahn for help with electron microscopy, and Susanne Gebhard (University of Bath) for kindly providing the *luxABCDE*-containing vector pAH328.

D.S., M.D., and C.N.R. conceived the study and designed the experiments. C.N.R., M.W., T.B., and J.K. carried out the transcriptome analysis; M.D. and R.U. designed the vectors for the bioluminescence reporter assay; M.D. generated plasmids and carried out the promoter evaluation. M.D. designed the promoter mutagenesis experiment; M.D. and L.B. generated and analyzed the promoter knockout mutants. M.D. analyzed the sequence conservation. M.D., C.N.R., and D.S. wrote the manuscript. All authors read and approved the final manuscript.

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
