## [Reviewer comments · mSystems]

The Complex Transcriptional Landscape of Magnetosome Gene Clusters in *Magnetospirillum gryphiswaldense*

Marina Dziuba, Cornelius Riese, Lion Borgert, Manuel Wittchen, Tobias Busche, Jörn Kalinowski, René Uebe, and Dirk Schüler

Corresponding Author(s): Dirk Schüler, University of Bayreuth

Review Timeline:

Submission Date:

July 14, 2021

Accepted:

August 11, 2021

Editor: Caroline Ajo-Franklin

Reviewer(s): The reviewers have opted to remain anonymous.

Transaction Report:

DOI: <https://doi.org/10.1128/mSystems.00893-21>

August 11, 2021

Prof. Dirk Schüler
University of Bayreuth
Microbiology
Bayreuth
Germany

Re: mSystems00893-21 (The Complex Transcriptional Landscape of Magnetosome Gene Clusters in *Magnetospirillum gryphiswaldense*)

Dear Prof. Dirk Schüler:

Your manuscript has been accepted, and I am forwarding it to the ASM Journals Department for publication. For your reference, ASM Journals' address is given below. Before it can be scheduled for publication, your manuscript will be checked by the mSystems senior production editor, Ellie Ghatineh, to make sure that all elements meet the technical requirements for publication. She will contact you if anything needs to be revised before copyediting and production can begin. Otherwise, you will be notified when your proofs are ready to be viewed.

As an open-access publication, mSystems receives no financial support from paid subscriptions and depends on authors' prompt payment of publication fees as soon as their articles are accepted. =

Publication Fees:

We recognize that the video files can become quite large, and so to avoid quality loss ASM suggests sending the video file via <https://www.wetransfer.com/>. When you have a final version of the video and the still ready to share, please send it to Ellie Ghatineh at eghatineh@asmusa.org.

Sincerely,

Caroline Ajo-Franklin
Editor, mSystems

Journals Department
Table S3: Accept
Fig. S3: Accept
Table S4: Accept
Fig. S1: Accept
Fig. S2: Accept
Table S1: Accept
Table S2: Accept
Fig. S4: Accept